# Medical Support for Space Missions: The Case of the SIRIUS Project

Stefania Fedyay *, Arslan Niiazov [ID], Sergey Ponomarev [ID], Aleksei Polyakov, Mark Belakovskiy and Oleg Orlov

Institute of Biomedical Problems (IBMP), Russian Academy of Sciences, Khoroshevskoye Shosse 76A, 123007 Moscow, Russia; arn1708@yandex.ru (A.N.); dr.grey@bk.ru (S.P.); sfedyai@yandex.ru (A.P.); msbelakovskiy@gmail.com (M.B.); isolationproject@bk.ru (O.O.)
* Correspondence: fed.stefany@gmail.com; Tel.: +7-(98)-51323538

**Abstract:** Medical support is one of the essential safety conditions for isolation or confinement experiments, as it enables the timely arrangement of actions to preserve the health of crew members and volunteers. Such analog experiments allow the testing of prospective medical technologies and methods for health support in long-term space missions and on-planet stations. In the current paper, we report the results of the medical control within the medical support system of the two model isolation experiments of the SIRIUS series, lasting for 4 and 8 months, respectively. The results indicate the prevalence of headache complaints, skin inflammatory reactions, and sleep disturbance during the longer confinement experiment. In addition, signs of vitamin D deficiency were revealed in 10 of the 12 objects. The data exchange with the scientific branch of the experiments provides for the in-time detection of early symptoms of disease, using samples of blood, urine, saliva, epithelia, etc. However, the issues of medical data confidence and, subsequently, of the crew members' compliance with the medical staff, become pointed. In general, the work demonstrates the expediency of the investigations, including the data collection and analysis of the medical control indicators in further experiments, for the optimization of the medical support of both the analogous research projects and the development of the recommendations for the medical support of small autonomous groups, such as manned space missions.

**Keywords:** isolation/confinement; medical support; model experiments; manned space missions; long-term space flights





## 1. Introduction

Over the last 60 years, humanity has successfully explored the Earth's orbital space, and presently, space researchers are focusing on the Moon. There are plans for small crews to land on the Moon surface and to build lunar orbital stations and moon bases for further work on engineering, medical, biological, and other technologies and methods to be used in long-term interplanetary missions [1].

The planning of the lunar missions is challenging and necessitates the revision of the existing bio-medical recommendations. The solution relies on the conduction of Earth-based analogous experiments, which model the main factors of a long-term space flight [2]. They involve anti-orthostatic hypokinesia [3–5], "dry" immersion [6,7], hypo-magnetic environment [8], simulation of gravity with the help of a centrifuge [9–12], parabolic flights [13], isolation experiments [14–17], and the confinement experiments in the analogous conditions of Arctic and Antarctic stations [18–21]. However, only the isolation experiments can provide the simulation of an autonomous operation of a small group, such as a crew of a spacecraft, and this is essential for the development of the medical support programs of the prolonged space flights.

## 1.1. The SIRIUS Projects

Since 2017, Russia has conducted the series of the SIRIUS (Scientific International Research in the Unique Terrestrial Station) isolation experiments modelling the conditions of a space flight. The project partners with the Institute of bio-medical problems (IBMP, Moscow, Russia) and the Human research program of National Aeronautics and Space Administration (NASA HRP, Houston, TX, USA) and aims at the investigation of the human body's adaptation to the conditions of isolation in a hermetically sealed chamber within an artificial environment. The SIRIUS experiments simulate the elements of interplanetary space flights of various durations, allowing the realization and improvement of the medical support program of real future missions beyond the Earth's orbit. The mission scenario consists of the following steps: "the flight" to the lunar orbital station with the subsequent "landing" on the satellite and "the return" to Earth. The international crew of the SIRIUS project includes six members, who perform a wide scientific program similarly to the crews of the International space station (ISS), and the results serve as a basis for the development of the bio-medical support of the upcoming interplanetary missions. SIRIUS projects are held in the ground experimental complex NEK in IBMP. NEK consists of four hermetically sealed facilities (with volumes of 50 $m^3$, 100 $m^3$, 150 $m^3$, and 250 $m^3$) equipped with autonomous life support systems (ventilation and air conditioning, water supply, sewerage, power supply, telecommunication, video control, firefighting system, system for the maintenance of the atmosphere gaseous content, temperature, and humidity). This allows for the isolation of the crew from the environment for the given duration of the experiment. At the preparation stage, the experimental complex is supplied with the means to provide first aid, as well as the medical units, and the equipment for medical control.

The programs of the experiments were approved by the Biomedical ethics committee of the IBMP of the physiology section of the Russian Committee for Bioethics under the Commission of the Russian Federation for UNESCO and by the NASA Committee for the Protection of Human Subjects, Human Research Multilateral Review Board.

## 1.2. Medical Support in the SIRIUS Isolation Experiments

There are three main stages in the isolation experiments simulating the space flight factors: pre-experimental, experimental, and post-experimental. A preparatory period may also be distinguished, during which a scientific program of an experiment is approved, the agreements between the participating countries are completed, the primary medical documentation is prepared and approved (e.g., test-volunteers' insurance; contracts with medical institutions for consultation, medical care, rehabilitation, or first aid; medical experimental rules; etc.). Moreover, the volunteers for the experiment are selected in the preparatory stage. The medical selection consists of the following steps:

1. The consideration of the primary medical documents of the test-volunteers;
2. The psychological testing of the volunteers (conducted by the specialists of the IBMP);
3. The medical examination of the volunteers by the recruited medical institutions (the list of the examinations is approved in advance for each of the experiments);
4. The session of the commission of medical experts of the IBMP on the results of the medical examinations. The commission decides whether a volunteer is selected for the project and nominated to the SIRIUS crewmembers.

The medical support of an experiment is held by a medical support group headed by a chief doctor of the SIRIUS project in the IBMP. The medical support group consists of the chief doctor, his/her deputy, and the doctors on duty. The doctors on duty head the medical crews on duty and are, in their turn, under the chief doctor. Besides the doctors on duty, during different stages of the experiments, the medical crews may include the laboratory assistants, engineers, and technicians, who are appointed to the posts by the competent staff after being attested by the commission (the knowledge of the equipment operation rules, protection of labor, regulations and instructions, documents on experimental procedures, etc., is tested). Only the certified and accredited specialists with higher education, who passed a probation period as laboratory assistants in other experiments in the IBMP, can be

appointed as doctors on duty. The engineers and technicians are under the chief engineer of the experiment.

The medical support of SIRIUS experiments rests on the principles of the evidence-based medicine. The medical equipment of the medical crews on duty is conducted in accordance with the local documents. However, as the local documents may fail to account for all the particular features of the experimental isolation studies, the medical equipment is also based on the experience of the medical support of other similar experiments and real space flights.

**The pre-experimental (or baseline) period** starts with the order of the IBMP director on the beginning of the background investigations and lasts up to the set of the isolation. All the selected candidates sign the informed consent for the participation in the experimental studies, then the testers (a tester is a volunteer participant of the experimental studies) are acquainted with the conditions of the experiment and the particular investigations, which are performed in popular written form, so that the volunteers have a clear idea of the procedures to be conducted and are aware of the associated risks.

At this stage, the volunteers use the laboratory and scientific equipment, explore the technical support of the experimental facility, undergo the preparation for isolation, and conduct those investigation methods, in the programs of which the background data collection is planned.

During the whole background period, all the dynamic and invasive procedures are accompanied by the doctors on duty, and the experiments with the 24 h observations are controlled by the crews on duty.

For the prevention of infectious diseases, the volunteers are held for observation for 5–7 days before the set of the isolation. During this period, the testers are accommodated in a special facility within the IBMP and are restricted to go outside. A limited number of researchers, whose experimental methods cannot be conducted remotely, as well as the staff providing the experiment, are allowed to visit the testers. At this stage, the medical crews, consisting of a doctor and a laboratory assistant, are on 24 h duty, and the medical examination is conducted on a daily basis. As a rule, at this time the session of the mandatory commission of the IBMP is held, and the final decision on the staff of the main crew and the reserved one is made. In 2021, the severe epidemiological situation caused by the COVID-19 pandemic made the observation period increase to 21 days, and the counter-epidemic preventive measures were significantly enforced.

**The experimental (or isolation) period** starts with the SIRIUS crew batting down the hatches of the ground experimental complex NEK. The mission scenario and the conditions of the experiment exclude a direct contact between the medical support group and the crew. To minimize the risks, one of the crewmembers plays the role of a crew doctor, who is under the chief doctor. The necessity of a physician within the hermetically sealed facility was repeatedly proved by the practical experience from the analogous experiments, so it is obligatory that at the preparatory stage of the experiment several candidates with medical education are selected. During the whole isolation period, the medical crews on duty, consisting of a doctor, a laboratory assistant, and a technician, provide the 24 h safety of the experiment. This is ensured by the video monitoring of all NEK modules and by the sensors of the life support systems. According to the rules of the experiment and the scenario of each SIRIUS mission, the crew and those on duty communicate only through the radiograms (electronic mail) or through the videoconference (video mail). Thus, the medical support of SIRIUS testers in the hermetically sealed chamber is realized through the crew doctor, who can directly communicate with the medical crew on duty and the chief doctor. If necessary, a telemedical consultation with any specialist from the medical institution recruited for the experiment is also possible. Moreover, all the crewmembers conduct their own daily medical examination (they measure body temperature, heart rate, arterial blood pressure, and weight) and put the data down into a specific register in the form of tables, to which the doctor on duty and the chief doctor have a direct access. The rules of the medical support imply an extended medical examination prescribed by the chief doctor based on

the testers' complaints. All the researchers and other people involved in conducting the experiment must inform the chief doctor (the chief doctor's deputy) or the doctor on duty of the aberrations in the health parameters of the crew members, if any are revealed. In case of need, even the scientific equipment can be used to provide aid for a tester. A private conference with a chief doctor is an essential part of the medical support. Their frequency depends on the duration of the experiment. Usually, they are conducted once a week or two by telephone, video communication, or in written form (according to the scenario of a mission). Medical recommendations are given to the crewmembers by the crew doctor after the chief doctor's approval. Should the chief doctor give direct prescriptions to a crewmember, he/she must inform the crew doctor. In addition, the medications consumed by the crewmembers are registered and strictly controlled. The chief doctor performs a weekly report of all medical cases to a chief executor of the SIRIUS mission.

In case of emergency or urgency (a tester's life and health are threatened), the crewmembers must contact the medical crew on duty without delay using the telephones placed in each of the NEK facilities. The chief doctor has a right to stop the experimental exposure for one, several, or all of the participants due to medical reasons.

**The post-experimental (recovery) period:** Upon the completion of the established isolation period, the crew of the "spacecraft" leave the NEK facilities. To bring the risks of illnesses during the acute adaptation to the environment to a minimum, after the isolation, the testers undergo a medical observation for 3–5 days, which is alike that in the pre-experimental period. The experiment is complete when all the scientific methods, which claim the collection of data in the recovery period, are conducted. Usually, this takes two weeks to one month, as the particular timeframes are established by the scientific curriculum of the experiment. During the whole post-experimental period, the chief doctor is still responsible for the medical support of the testers. While the testers stay at the IBMP, the duty of the medical crews proceed, and the doctor on duty accompanies the crewmembers on the procedures held in different medical institutions, if needed.

The stages of the SIRIUS missions are schematically presented in Figure 1.

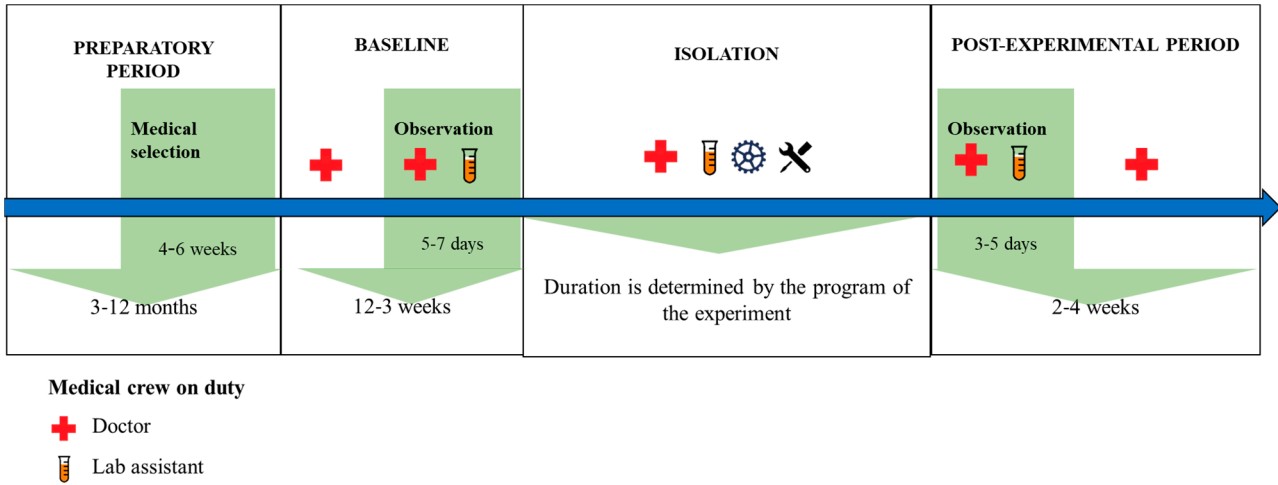

**Figure 1.** The scheme of the experimental stages of the SIRIUS missions.

## 2. Materials and Methods

In the current paper, we present the data from the two isolation experiments of the SIRIUS project which lasted for 4 and 8 months. In the 4-month (120 days) SIRIUS-19 experiment held in 2019, 6 participants aged 29–45, including 3 male and 3 female volunteers, took part. In the 8-month (240 days) SIRIUS-21 experiment held in 2021–2022, there were 6 testers aged 28–45, with 3 males and 3 females staying in the facility for

1 month. After this, one of the crewmembers left the facility, and there were 3 males and 2 females.

Here, the data from the medical control log-book, in which the complaints, diagnoses and prescribed medications are registered, were used. These data were analyzed by descriptive statistical methods using the Microsoft Office Excel software and a specially designed computer program based on Python 3.0 (Jupyter Notebook 6.4.8).

The programs of the experiments were approved by the Biomedical ethics committee of the IBMP of the physiology section of the Russian Committee for Bioethics under the Commission of the Russian Federation for UNESCO and by the NASA Committee for the Protection of Human Subjects Human Research Multilateral Review Board (protocol № 501 from 18.02.2019 for SIRIUS-19; protocol № 539 from 17.03.2020 for SIRIUS-21).

For confidentiality purposes, the sex, age, and role in the crew are omitted in the current paper.

## 3. Results

We analyzed all the medical cases identified in the SIRIUS-19 and SIRIUS-21 isolation experiments. By the term 'medical case', we mean a disease case that meets the criteria established for identifying a specific disease according to the 10th Revision of the International Classification of Diseases (ICD-10). The term 'complaints' is used to focus attention on symptoms or reported issues communicated by the crew members themselves. For confidentiality purposes, the data are represented as total values of the crews in Figure 2.

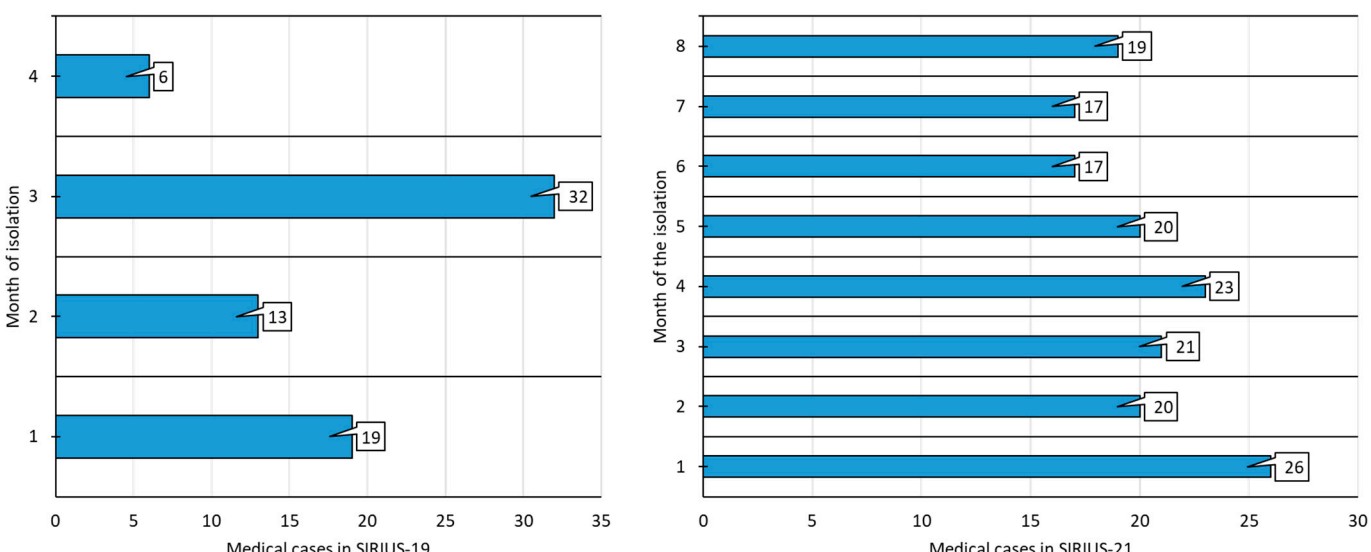

**Figure 2.** The number of clinical cases in the SIRIUS-19 and SIRIUS-21 isolation experiments.

In the 4-month experiment, out of the 70 clinically significant medical cases, 32 (46%) were related to the scientific methods. In the 8-month isolation, only 15 (9%) out of 163 cases were caused by the scientific experiments. Generally, they included skin reactions to electrodes manifested as itching and reddening, bruises and surface traumas, and injuries after venipunctures.

In the SIRIUS-19 experiment, 34 out of the 70 cases (49%) were based on subjective complaints, in the remaining 36 cases (51%), the objective changes were revealed (such as visual signs during medical examination by the doctor, laboratory or instrumental examination). They included five cases (7%) which reported the data of laboratory examinations. In the SIRIUS-21 experiment, 79 out of the 163 complaints (48%) were subjective, while 84 cases (51%) were supported by objective methods, including laboratory ones in 11 cases (7%).

The clinical cases were analyzed on the basis of syndromes and conditionally united into diagnostic groups according to the ICD-10 (Figure 3).

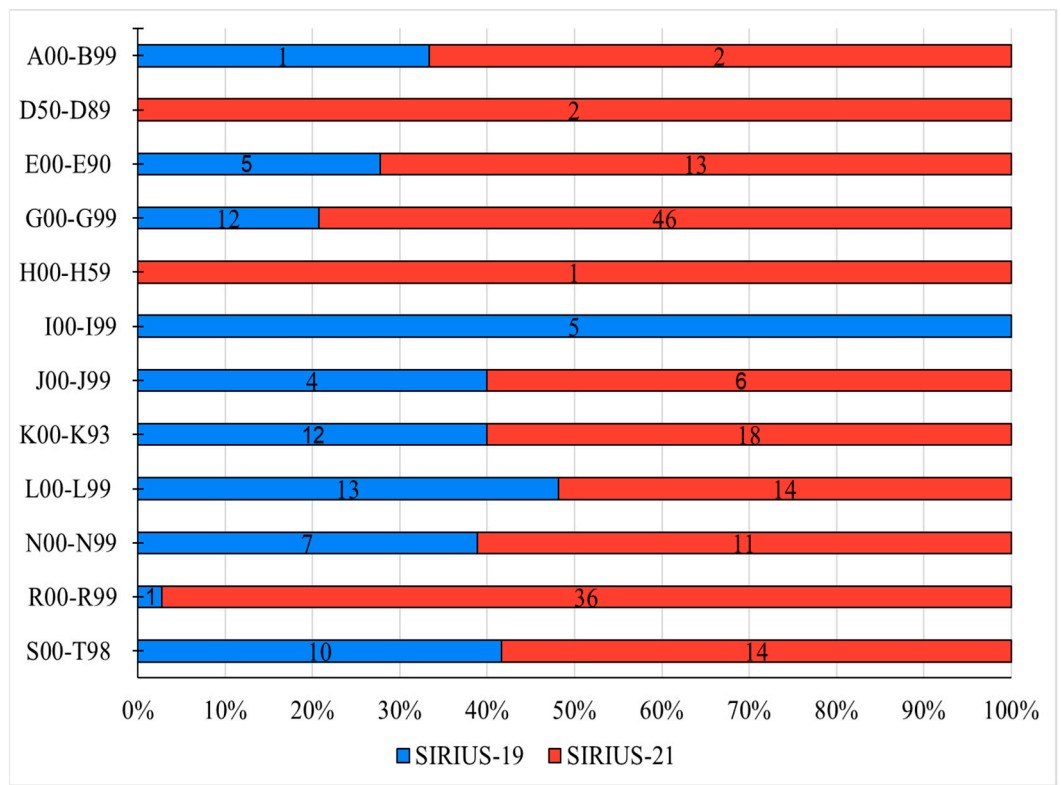

**Figure 3.** The revealed medical cases in the 120-day and 240-day isolations united into diagnostic groups. The codes for ICD-10 are listed in Appendix A Table A1.

The prevalent complaints in both of the experiments were skin conditions and headaches. In the longer isolation, the crew members demonstrated sleep disturbances during the whole exposure period. In general, the pathologies revealed in the two missions correlated (with the exception of sleep disorders). In both SIRIUS missions, some of the crew had rhinitis and nasopharyngitis during the first days of stay in the hermetically sealed chambers.

Rubric R00–R99 is of particular interest, as it includes symptoms and signs that could not be classified more specifically in other rubrics (e.g., R10–R14, R20–R21, R53, R55, R80–R82).

There were 2 of 70 (3%) in the SIRIUS-19 cases and 2 of 163 (1%) in the SIRIUS-21 cases of the direct risk of a pre-term leaving of the experiment for medical indications. In 1 case out of 163 (0.6%) in the SIRIUS-21 project, the tester was withdrawn from the experiment for medical reasons because of trauma. Moreover, there were two cases (one in the SIRIUS-19 and one in the SIRIUS-21) in which the crew members demonstrated dental problems, and in one case in the SIRIUS-19, there was a syncope state with a short-term unconsciousness.

Out of the 70 clinical cases in the 4-month isolation, 48 cases (69%) required medical intervention. In the 8-month isolation, the medical intervention was carried out in 91 clinically significant cases out of the 163 (56%). A total of 7 out of 70 cases (10%) in the SIRIUS-19 and 12 out of 163 cases (7%) in the SIRIUS-21 demanded corrections in the daily schedules. It is worth noting that these corrections did not affect the scientific programs. Medical manipulations were conducted in 11 cases out of 70 (16%) in the 4-month isolation and in 6 cases out of 163 (4%) in the 8-month isolation.

Medications were prescribed in 44 out of 70 cases (63%) in the SIRIUS-19 and in 81 out of 163 cases (50%) in the SIRIUS-21. The medications prescribed in both missions along with the putative frequently used ones are summarized in Table 1, and the summary is applied for the estimation of their supply during the planning of the upcoming isolation experiments.

**Table 1.** Medications prescribed in the SIRIUS-19 and SIRIUS-21 projects and the estimation of putative frequently used medications for the calculation of their supply in future experiments.

| Name of the Medication and Dosage | Average Number of Doses for a Person per Month of the Experiment | | Suppositional Average Number of Doses for a Person per Month of the Experiment |
|---|---|---|---|
| | SIRIUS-19 | SIRIUS-21 | |
| **Usage: orally** | | | |
| Ibuprofen, 400 mg | 0.92 | 1.29 | 1.10 |
| Melatonin, 3 mg | 0 | 1.23 | 0.61 |
| Amoxicillin + Clavulanic Acid, 875 mg + 125 mg | 0.88 | 0 | 0.44 |
| Valacyclovir, 500 mg | 0.50 | 0 | 0.25 |
| Paracetamol, 500 mg | 0 | 0.45 | 0.23 |
| Ketoprofen, 100 mg | 0.17 | 0 | 0.08 |
| Polymethylsiloxane polyhydrate, 15 mg | 0.13 | 0 | 0.06 |
| Desloratadine, 5 mg | 0.08 | 0 | 0.04 |
| Simeticone, 40 mg | 0.08 | 0 | 0.04 |
| Cetirizine, 10 mg | 0 | 0.05 | 0.03 |
| Mebeverine, 200 mg | 0.04 | 0 | 0.02 |
| **Usage: topically** | | | |
| Saline solution, hypertonic | 7.08 | 1.63 | 4.35 |
| Acyclovir 5%, cream | 1.75 | 2.83 | 2.29 |
| Dimetindene 0.1%, gel | 3.00 | 0 | 1.50 |
| Hypromellose + Dextran (Natural Teardrop) eye drops | 0 | 2.30 | 1.15 |
| Benzidamine, 3 mg | 1.50 | 0.70 | 1.10 |
| Miramistin 0.01%, solution | 1.17 | 1.03 | 1.10 |
| Heparin + Dexpanthenol + Dimethyl Sulfoxide, gel | 0.25 | 1.80 | 1.02 |
| Bacitracin + Neomycin, 250 IU/g + 5000 IU/g, ointment | 0 | 1.75 | 0.88 |
| Ketoprofen 2.5%, gel | 0.00 | 1.33 | 0.81 |
| Allantoin + Heparin + Allii cepae squamae extract, gel | 0 | 1.40 | 0.70 |
| Hydrogen peroixide 3%, solution | 0.17 | 1.13 | 0.65 |
| Lidocaine + Chamomillae recutitae floridis extract, 20 mg/g + 185 mg/g, gel | 0.38 | 0.23 | 0.30 |
| Bacitracin + Neomycin + Polymyxin B, 400 mg/g + 3.5 mg/g + 5000 IU/g, ointment | 0 | 0.45 | 0.23 |
| Mometasone 0.1%, cream | 0 | 0.43 | 0.21 |
| Betamethasone + Gentamicin + Clotrimazole, 0.5 mg/g + 1 mg/g + 10 mg/g, ointment | 0 | 0.35 | 0.18 |
| Diclofenac 1%, gel | 0.25 | 0 | 0.13 |

Table 1. *Cont.*

| Name of the Medication and Dosage | Average Number of Doses for a Person per Month of the Experiment | | Suppositional Average Number of Doses for a Person per Month of the Experiment |
|---|---|---|---|
| | SIRIUS-19 | SIRIUS-21 | |
| Dioxomethyltetrahydropyrimidine + Chloramphenicol, 40 mg/g + 7.5 mg/g, ointment | 0.17 | 0 | 0.08 |
| Cromoglycate 20 mg/mL, eye drops | 0 | 0.05 | 0.03 |
| Body Skin Glue BF-6 | 0.04 | 0 | 0.02 |
| Dexpanthenol 5%, spray | 0.04 | 0 | 0.02 |
| **Usage: parenterally; intramuscular** | | | |
| Ketorolac, 60 mg | 0 | 0.02 | 0.01 |

In addition to the medications listed in Table 1, 5 crewmembers out of 6 in both of the projects were supplied with a daily dose of 7000 ME of vitamin D for 4–8 weeks due to the revealed deficiency. On average, the vitamin D deficiency was detected on day 66 of the experiment.

Additionally, in both experiments, participants were recommended to take a vitamin and mineral complex as a dietary supplement in a standard daily dose (a complex biological supplement that is not classified as a medication). It was advised to take one tablet per day, containing 13 essential vitamins and 8 minerals, in doses not exceeding the recommended upper limit of consumption. This supplementation demonstrated a positive effect, particularly in addressing signs of vitamin or micronutrient deficiencies, primarily related to skin conditions. The deficiency of vitamins and microelements was detected on day 48 of stay in the hermetically sealed chamber.

Among the oral medications, non-steroid anti-inflammatory drugs (ibuprofen) appeared to be predominantly prescribed. In the longer isolation experiment, the sleep disturbances were relatively frequently treated with melatonin.

## 4. Discussion

The study of the somatic and functional aberrations in the testers, as well as the estimation of the medical support system operation in the isolation experiments, are highly relevant in regard to the increased duration of similar missions and their relatedness to the medical support of real spaceflights.

Based on the review conducted, we propose the following definition of the medical support of the isolation experiments: it is a series of the organizationally, structurally, and functionally interconnected measures and actions of medical control (including diagnostics), precautions and therapy, which provide both the autonomous management of a volunteer's functional state and the monitoring by the ground medical services during the experiment. In addition to health preservation, the medical support system of the isolation experiments also aims at the improvement of the quality of the scientific data obtained during the experiments and the methods of the medical support of space missions.

The analyses of the real medical cases appeared in the course of the SIRIUS projects, and the estimation of the appropriateness of the conducted medical interventions allow one not only to foresee the risks of the clinical cases in further experiments and, subsequently, in the interplanetary missions, but also to develop the effective countermeasures.

In the isolation experiments described in our study, dermatological pathology, headaches, and trauma and/or injuries appeared to be most frequent. In the 4-month mission, a relatively high percentage of gastrointestinal disorders was registered, while the crew members of the 8-month missions claimed sleep disturbances. According to the analysis of the incidences on the "Mir" space station, the cosmonauts suffered mostly from insom-

nia, headaches, and traumatic or thermic skin injuries, as well as skin allergy caused by electrodes [22].

Onboard ISS, the crewmembers receive therapy for the correction of specific disorders, caused by the spaceflight conditions, and for some common non-specific diseases [23]. The adverse effects of space flights generally include movement disorders and sleep disturbances. Among the usual complaints, headaches, congestive states, allergy, and respiratory infectious manifestations are predominantly registered. According to the published data, on ISS, such as in the 240-day isolation experiment, the medications for sleep corrections are frequently taken. Sleep–wake cycle disruptions were also observed in the unique 520-day isolation experiment, MARS-500, where the authors of the sleep study demonstrated that not only the quality of sleep but also the quality of wakefulness of the volunteers was affected [24]. The fact that no sleep disorders were detected in the 120-day isolation experiment was surprising, though it needs a particular analysis, since the conditions in the hermetically sealed chamber, including light, noise, intensity of the scientific program, etc., were the same in both SIRIUS projects.

Due to the timely interaction of the doctors involved in the experiments and the research groups in charge, with the investigation of metabolic changes, the vitamin D deficiency in 10 volunteers was detected and subsequently corrected. The problem of the vitamin D deficiency is still acute onboard the International space station, for this substance affects many body processes, including the development of osteoporosis [25].

Another burning issue is the confidentiality of the medical data in the scientific experiments. The confidentiality and ethical standards must be strictly followed. The awareness of a volunteer that the information on his/her health is transferred to a third-party negatively affects the confidence between the volunteer and the specialists of the medical support. This may lead to the data formalization or dissimulation, late consultation with the doctors (despite the written informed consent about the obligations) and threaten the volunteer's health and the mission. It is important to consider that the doctors in the experiment cannot be responsible for the crew members' health in case they conceal their health problems, inform the medical staff about their state in an untimely manner, or perform self-treatment or self-diagnostics and take therapy from an exterior specialist without consultation with an IBMP doctor. Manned astronautics abounds such examples. Confiding relations between the chief doctor, the crew doctor, and the crewmember defines the efficiency of pathology diagnostics, a volunteer's compliancy, and, finally, the successful mission completion. At the same time, a tight cooperation between the medical supply service and the scientific staff of the experiment is required for the minimization of the effect of clinical cases and medical interventions on the realization of the experimental program and results of the research. In this regard, there is an issue of the transfer of private medical information, in particular, of the medications prescribed or of the clinical states of a volunteer, to a researcher, whose experimental results may be affected by the situation. For instance, the immunological studies imply the comparison of laboratory analyses and clinical manifestations for revealing the early signs of diseases, primarily, those connected with immunity malfunctioning (reactivation of latent intracellular infections or opportunistic pathogens, allergy, or autoimmune reactions). Moreover, the data of medical monitoring and of clinical disease manifestations (if any) are extremely important for the establishment of the individual variability ranges of the immune effectors' fluctuations. The volunteers' health parameters together with the medications prescribed are also vital for the objective estimation of the immune countermeasures tests, as the results may be affected by the remedies [26,27].

Apparently, there is a complicated situation when, on the one hand, the researchers should be informed of the testers' medical state, and on the other hand, the privacy of this information should be preserved. In the SIRIUS projects, the fact that the medical data may be transferred to a researcher upon a valid request and agreement of the experiment's management is explained to the volunteers.

## 5. Conclusions

1.  The medical support of the experimental projects, as well as of small autonomous groups, should take into account their specificity and basis on the principles of the demonstrative medicine with the application of the latest progress of translational medicine and integrative physiology modified in accordance with the data analysis on the medical cases. The further improvement of the first medical aid, including that during traumas, along with the adaptation of the protocols and methods to the isolation of small groups, as well as the appropriate staff training, are expedient.

2.  The experience of the medical experiments confirms the significance of the medical support service and, particularly, of the crews on duty, both for the safety of the volunteers and the scientific program realization. The operation of the medical support should be based on the transparency and structure hierarchy, and exclude the conflict of interests between the volunteers' safety and the obtaining of the scientific data. The safety comes first, as it defines the quality of the data.

3.  The doctor on duty should control all the scientific experimental procedures during the pre-experimental and the post-experimental periods, as they are characterized by an increased risk of injuries (dynamic physical tests and blood sampling). Moreover, the volunteers should be accompanied by the doctor on duty to other institutions, if necessary.

4.  In the isolation experiments lasting more than 1 month, vitamin D uptake is recommended, starting as early as the pre-experimental period.

5.  In the isolation experiments lasting more than 1 month (which imply primarily canned and dried food), vitamin and mineral complex uptake is recommended, starting from the pre-exposure period as a dietary supplement.

6.  The issue of the timely consideration of the subjective feelings of the testers, especially in the cases of pain syndrome, for the selection of an efficient medical treatment, remains open.

7.  The further medical data collection and analysis in the upcoming experiments is expedient for the optimization of medical support in analogous research experiments and for the development of the recommendations for the medical support of small autonomous groups, including distant manned space missions.

**Author Contributions:** Conceptualization, O.O., S.F. and A.N.; methodology, S.F. and A.N.; software, A.N.; formal analysis, A.N. and S.F.; investigation, A.N. and S.F.; resources, O.O. and S.P.; data curation, A.N. and S.F.; writing—original draft preparation, S.F.; writing—review and editing, S.F.; visualization, S.F.; supervision, A.P. and S.P.; project administration, O.O. and M.B.; funding acquisition, S.P. All authors have read and agreed to the published version of the manuscript.

**Funding:** The study was supported by the Ministry of Science and Higher Education of the Russian Federation under agreement № 075-15-2022-298 from 18 April 2022 about the grant in the form of subsidy from the federal budget to provide government support for the creation and development of a world-class research center, the "Pavlov Center for Integrative Physiology to Medicine, High-tech Healthcare and Stress Tolerance Technologies.

**Data Availability Statement:** Data are contained within the article.

**Conflicts of Interest:** The authors declare no conflict of interest.

## Appendix A

**Table A1.** ICD-10 codes.

| Code | Diagnostic Group |
| --- | --- |
| A00–B99 | Infectious and parasitic diseases |
| D50–D89 | Diseases of the blood and blood-forming organs |
| E00–E90 | Endocrine, nutritional, and metabolic diseases |
| G00–G99 | Diseases of the nervous system |
| H00–H59 | Diseases of the eye and adnexa |
| I00–I99 | Diseases of the circulatory system |
| J00–J99 | Diseases of the respiratory system |
| K00–K93 | Diseases of the digestive system |
| L00–L99 | Diseases of the skin and subcutaneous tissue |
| N00–N99 | Diseases of the genitourinary system |
| R00–R99 | Symptoms, signs, and abnormal clinical and laboratory findings, not elsewhere classified |
| S00–T98 | Injury, poisoning, and certain other consequences of external causes |

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
