# Peer review of "Medical Support for Space Missions: The Case of the SIRIUS Project"

_aerospace, doi:10.3390/aerospace10060518_

Round 1

Reviewer 1 Report

This is an interesting paper and should be published. The results section begins with the statement "We analyzed all the medical cases, ..." yet what constitutes a "medical case" is never defined. Is it an observation by the medical team and/or a symptom or sign reported by a crew member? Sometimes the words "cases" and "complaints" are used interchangeably. Without proper definition and clarity of what a case is, the results section is not as strong as it should be.

Sleep disturbances in the longer SIRIUS-21 experiment are mentioned several times, as is the importance of isolation experiments in assessing the operation of a small group in a simulated spaceflight mission. Reference to the findings of Basner et al., PNAS, 2013, from the MARS 500 520-day isolation study in the NEK should be included.

The quality of the English needs to be improved. The first sentence for the reader in the abstract contains the phrase "as it enables to timely arrange the actions." The is not grammatically correct. Perhaps it should read "as it enables the timely arrangement of actions." The paper requires careful editing prior to publication.

Author Response

Dear Reviewer,

Thank you for your valuable feedback and contribution to improving our work. We sincerely appreciate your interest in our study and your recommendation for publication.

In response to your comment about the Results section, we agree that a clearer definition of the term "medical case" is necessary. We have included it to our article as well as a definition of the term “complaints”.

We also agree with your recommendation to include a reference to the study by Basner et al. (PNAS, 2013) from the MARS 500 study. We have incorporated a reference to this study in the relevant section of our work.

The importance of thorough editing and proofreading of the article prior to publication is duly recognized. We have engaged a native English speaker and a specialist in the related field to review and address any language errors. We are confident that their expertise will significantly contribute to improving the overall quality of the English language in the final version of the article.

With the best regards, Stefania O. Fedyay also on behalf of all co-authors,

Reviewer 2 Report

The authors Fedyay et al. present results from two campaigns of the SIRIUS Isolation study at the NEK isolation facility in Moscow, Russia. Since incidence data of clinical problems in space flight and space flight analogues are very rarely accessible to the scientific community, this paper is of high significance. The section in the discussion where confidentiality is discussed is a well known problem and I would like to thank the authors that they addressed and discussed this problem in their paper.

I have only two minor questions/comments:

Can you elaborate on the section R00-R99, since there are many cases and it would be interesting to know what symptoms/diseases exactly where present. If they refer to cases you discuss already, than simply point this out in the text.

Can you disclose what "mineral complex in a standard daily dose as a dietary supplement" exactly was? 

Thank you.

Author Response

Dear reviewer,

We would like to express our gratitude for your feedback and valuable comments on our work. We highly appreciate your recognition of the significance of our study and we are delighted to know that you find our paper to be of great importance. Additionally, we would like to thank you for acknowledging the section in the discussion where confidentiality is addressed.

In response to your two questions/comments, we are pleased to provide further clarification:

  1. The R00-R99 section indeed corresponds to a significant portion of the International Statistical Classification of Diseases and Related Health Problems, Tenth Revision (ICD-10). We have incorporated appropriate clarifications in the article to ensure better understanding.
  2. We have provided additional information in the article about the "mineral complex in a standard daily dose as a dietary supplement." Unfortunately, due to its nature as a comprehensive biological supplement that is not classified as a medicinal product, it is only commercially referred to by its trade name. This supplement is recommended for use in the Russian segment of the International Space Station (ISS).

With the best regards,

Stefania O. Fedyay also on behalf of all co-authors.